# Primary Neuroendocrine Neoplasms of the Breast: Case Series and Literature Review

**DOI:** 10.3390/cancers12030733

**Published:** 2020-03-20

**Authors:** Burcin Özdirik, Antonin Kayser, Andrea Ullrich, Lynn J. Savic, Markus Reiss, Frank Tacke, Bertram Wiedenmann, Henning Jann, Christoph Roderburg

**Affiliations:** 1Department of Gastroenterology/Hepatology, Charité University Medical Center Berlin, Campus Virchow Klinikum and Charité Mitte, Augustenburger Platz 1, 13353 Berlin, Germany; burcin.oezdirik@charite.de (B.Ö.); antonin.kayser@t-online.de (A.K.); markus.reiss@charite.de (M.R.); frank.tacke@charite.de (F.T.); bertram.wiedenmann@charite.de (B.W.); henning.jann@charite.de (H.J.); 2Department of Pathology, Charité University Medicine Berlin, Charitéplatz 1, 10117 Berlin, Germany; andrea.ullrich@charite.de; 3Department of Diagnostic and Interventional Radiology, Charité University Medicine Berlin, Augustenburgerplatz 1, 13353 Berlin, Germany; lynn-jeanette.savic@charite.de

**Keywords:** neuroendocrine carcinoma of the breast, small cell carcinoma of the breast, case series, review, diagnostics, histology, management

## Abstract

Primary neuroendocrine carcinoma of the breast (NECB) as defined by the World Health Organization (WHO) in 2012 is a rare, but possibly under-diagnosed entity. It is heterogeneous as it entails a wide spectrum of diseases comprising both well-differentiated neuroendocrine tumors of the breast as well as highly aggressive small cell carcinomas. Retrospective screening of hospital charts of 612 patients (2008–2019) from our specialized outpatient unit for neuroendocrine neoplasia revealed five patients diagnosed with NECB. Given the low prevalence of these malignancies, correct diagnosis remains a challenge that requires an interdisciplinary approach. Specifically, NECB may be misclassified as carcinoma of the breast with neuroendocrine differentiation, carcinomas of the breast of no special type/invasive ductal carcinoma, or a metastasis to the breast. Therefore, this study presents multifaceted characteristics as well as the clinical course of these patients and discusses the five cases from our institution in the context of available literature.

## 1. Introduction

Neuroendocrine neoplasia (NEN) are a rare, heterogeneous group of tumors that originate from the diffuse endocrine system with variable clinical behavior depending on the differentiation of the tumor. According to the WHO classification, NEN can be stratified based on their histological differentiation into low- (grade 1; G1), intermediate- (grade 2; G2), and high-grade (grade 3; G3) neuroendocrine tumors (NET) and poorly differentiated neuroendocrine carcinoma (NEC), featuring a highly elevated Ki67 proliferative index and/or mitotic rate [1]. Well-differentiated NET (G1, G2) typically have a low proliferative index and retain high expression of somatostatin receptors (SSTR). Advanced well-differentiated NET that are ineligible for resection are managed in most cases with somatostatin analogs (SSAs), such as octreotide and lantreotide, peptide receptor radionuclide therapy and targeted agents, such as everolimus and sunitinib [2,3,4,5]. In contrast, poorly differentiated carcinomas are associated with rapid progression and a poor long-term prognosis. In most cases, these patients are treated with systemic cytotoxic chemotherapy. Neuroendocrine neoplasia may occur in almost all organ systems. In most cases NEN occur within the gastroenteropancreatic system (70% of all cases) and the bronchopulmonary system (25%) [6]. Examples of rare primaries are thyroid gland (8.6%) [7], skin (5%) [7], bladder (0.35–1%) [8] and larynx (0.23%) [9]. Mammary origin accounts for less than 1% among neuroendocrine tumors [10,11,12,13,14]. Their incidence among breast cancer has been reported to range from 0.1% to 5% [13,15,16,17]. These tumors are thought to arise from endocrine differentiation of breast carcinoma rather than from pre-existing endocrine cells with malignant transformation [18]. Considering the low frequency of neuroendocrine carcinoma of the breast (NECB), there is limited knowledge on the clinical presentation and management of this disease. Within this report, we highlight the clinical course of five patients with NECB presenting between 2008 and 2019 within our outpatient unit.

## 2. Case Reports

We present a series of five patients with histologically confirmed diagnosis with primary neuroendocrine neoplasm of the breast who were treated in our outpatient clinic for neuroendocrine tumors. Patient characteristics (including key diagnostics and therapies) are provided in Table 1, Table 2 and Table 3.

### 2.1. Case 1

A 73-year-old female patient was diagnosed in February 2019 with a well-differentiated, non-functional SSTR-positive NET G2 of the breast. The patient initially presented with a non-painful lesion of the breast detected during routine gynecological examination. The patient did not report any episodes of flushing or diarrhea. Physical examination revealed a palpable mass with overlying skin thickening on the left breast. Subsequent mammography demonstrated a suspicious lesion (10 mm diameter) at 7 o’clock position of the left breast, which turned out as BI-RADS (Breast Imaging-Reporting and Data System) 4b/5 in ultrasound, leading to a punch biopsy revealing a well-differentiated neuroendocrine tumor. Multi-slice computed tomography (CT) of chest and abdomen and bone-scintigraphy were performed, however no distant metastases were found. The tumor demonstrated high trace uptake in DOTATOC-PET/CT. Both serum chromogranin A levels and urinary 5-hydroxyindoleacetic acid (5-HIAA) excretion were normal.

Two weeks later the patient underwent partial mastectomy including lymphadenectomy. Pathological work-up of the resected tumor was performed by a reference pathologist and revealed a well-differentiated neuroendocrine tumor of the breast and one lymph node metastasis (1/8). On an immunohistological level, the tumor showed strong expression of synaptophysin (Figure 1a), SSTR2A (Figure 1b) chromogranin, nuclear hormone receptors (100% estrogen receptors (Figure 1c), 40% progesterone receptors), and GATA 3 (Figure 1d) supporting the hypothesis of a primary tumor of the breast rather than a metastasis of another (not detected) tumor. No expression of specific transcription factors such as TTF-1, CDX2, and Islet1 was found. Ki-67 index was 6%.

Clinically the patient was free of any symptoms of carcinoid syndrome. After surgery the patient refused any adjuvant treatment and only accepted regular follow-up imaging. One year after initial diagnosis, the patient is in a very well condition, still showing complete remission.

### 2.2. Case 2

A 53-year-old female patient was diagnosed with a well-differentiated G2 NET of the breast in November 2017. The patient initially presented with painful mastitis in the right breast to a gynecology outpatient clinic in July 2017. Ultrasound imaging suggested an abscess formation of the right breast at 6 o’clock, mammography was not performed due to inacceptable pain. Suspecting a bacterial mastitis, antibiotic treatment was initiated. Concomitant weight loss (4 kg within 6 months) was interpreted as stress related. In the following weeks, the patient’s condition improved only slightly, leading to a second presentation at the outpatient clinic. Physical examination revealed a previously unseen nipple retraction. A subsequently performed mammography revealed an irregular, star-like tumorous lesion. Further radiological work-up (CT chest/abdomen) showed no other sites of involvement. Core needle biopsy of the lesion was compatible with the diagnosis of a well-differentiated G2 neuroendocrine tumor. Serum chromogranin A and urinary 5-hydroxyindoleacetic acid (5-HIAA) excretion were not elevated.

A right-sided mastectomy including lymphadenectomy was performed in December 2017. No lymph node metastases were detected. Immunohistochemical staining revealed strong expression of synaptophysin (Figure 2a), ultimately proving the neuroendocrine differentiation, and strong expression of hormone receptors (ERR >80% (Figure 2b), PRR >90% (Figure 2c)). Staining for membranous HER2neu, CDX2, TTF1, and Islet1 was negative, while GATA3 (Figure 2d) expression hinted towards a mammary origin.

The proliferation rate according to Ki-67 was up to 10%. Curative resection was followed by 15 cycles of adjuvant radiotherapy. One month later antihormonal treatment with Tamoxifen was initiated. Two years later the therapy is still being continued and the patient is in good general health, free from tumor recurrence.

### 2.3. Case 3

A 49-year-old female was diagnosed with moderately-differentiated, non-functional G2 NET of the breast in March 2009. Routine mammography, performed due to a family history of breast cancer, revealed a malignant lesion in the upper outer quadrant of the right breast at 10–11 o’clock position. A subsequently performed core needle biopsy showed a ductal carcinoma in situ (DCIS) (Figure 3a) with positive hormone receptor expression. Further clinical work-up including multi-slice CT imaging was conducted but did not reveal any metastases. Somatostatin receptor scintigraphy (SRS) or PET-computed tomography was not performed at that time, since NET was not considered.

Two weeks later the patient was admitted to partial mastectomy. In further pathological work-up of the resected tumor, no lymph-node metastases were detected. Immunohistochemical staining showed a well-differentiated neuroendocrine tumor with expression of neuroendocrine markers such as synaptophysin (Figure 3b) and chromogranin. Nuclear estrogen and progesterone receptors (Figure 3c,d) and membranous Her2Neu expression were positive.

Two months after surgery, adjuvant radiotherapy was initiated. After adjuvant radiotherapy the patient discussed antihormonal treatment with her primary care gynecologist and rejected further treatment with tamoxifen, since there were no clear guidelines on antihormonal therapy for NEN. At present, the patient is still in complete remission and physically well.

### 2.4. Case 4

A 78-year-old female patient was diagnosed with small cell neuroendocrine carcinoma of the breast and synchronous liver, bone, pleural metastases and peritoneal carcinomatosis in May 2018. The patients’ family history was positive for small cell lung cancer (brother) and leukemia (father). Four months before initial diagnosis the patient complained of pain in the legs and spine. Further clinical work-up (multi-slice CT imaging, mammography) revealed multiple suspicious masses in both breasts as well as liver, bone, lymph node metastases and peritoneal carcinomatosis (Figure 4a–c). Bronchoscopy, upper endoscopy, and colonoscopy did not provide any evidence of another possible primary tumor.

A core needle biopsy from the right breast was taken, showing a small cell carcinoma of the breast. Immunohistochemical staining revealed positive expression of synaptophysin and E-cadherin, which is expressed in primary breast tumors [19,20]. Nuclear PR expression was positive in 15% of tumor cells. ER, HER2neu, CD3, CD20 were negative. Ki-67 was 40%. Serum chromogranin A and urinary 5-hydroxyindoleacetic acid (5-HIAA) excretion were normal.

Systemic chemotherapy was initiated using a chemotherapeutic regimen based on Carboplatin and Etoposide in June 2018. Moreover, the patient was treated with zoledronic acid. After administration of six cycles of chemotherapy in October 2018, the patient was referred to our outpatient clinic. CT imaging revealed stable disease and the patient received further chemotherapy, but died shortly after, from pulmonary embolism in February 2019.

### 2.5. Case 5

A 67-year-old female patient was diagnosed with a NEC G3 of the right mamma in August 2016. Routine mammography revealed a suspicious lesion at 12 o’clock position of the right breast. Subsequently conducted multi-slice computed tomography confirmed diagnosis of a suspicious breast lesion and detected right-sided hilar lymphadenopathy and multiple liver lesions (Figure 5a,b).

Further clinical work-up including bronchoscopy, upper endoscopy, capsule endoscopy, and colonoscopy did not reveal further pathological findings, not providing any evidence of another possible primary tumor. Therefore, the breast lesion was considered the most probable primary tumor site. Immunohistochemical analysis of a biopsy taken from the suspicious lesion in the breast showed a neuroendocrine small cell carcinoma (Figure 6a) with strong expression of synaptophysin (Figure 6b) and slightly weaker expression of chromogranin (Figure 6c) as well as positivity for pan-cytokeratin. Sporadic nuclear positivity for GATA3 (Figure 6d) was detected in tumor cells, while TTF1 and CDX2 were negative. Ki-67 was up to 80%.

Since the liver lesions showed typical patterns in contrast-enhanced ultrasound imaging (CEUS), biopsy was not considered necessary.

In this setting, chemotherapy with Cisplatin and Etoposide was initiated. CT imaging in November 2016 revealed stable disease. Although treatment was initially well tolerated, therapy had to be discontinued due to renal impairment after five cycles of chemotherapy in December 2016. Further imaging in March 2017 indicated disease progression with size increase of the primary tumor and liver metastases as well as the appearance of new lesions in the lung and bones. Therefore, a second line treatment with Doxorubicin, Cyclophosphamide (750 mg/m^2^), and Vincristine (0.4 mg/m^2^) was initiated. After administration of two cycles chemotherapy, laboratory testing revealed neutropenia stage IV. Initially, the patient did not present any clinical complications from chemotherapy, but died three months later, in July 2017 from multiorgan failure.

## 3. Review of the Literature and Discussion

### 3.1. Summary of the Case Reports

We report a series of five patients with histologically confirmed diagnosis of NECB who have been followed at our institution. According to the 2012 WHO classification, three patients (cases 1–3) suffered from a well/moderately differentiated NECB and two patients (cases 4 and 5) suffered from a small cell carcinoma of the breast.

Two out of three patients with well/moderately differentiated NECB (cases 1–3) presented with clinical symptoms (pain, erythema, skin and nipple retraction). Clinical work-up included mammography, ultrasound, CT-imaging, and punch biopsy in all patients. Interestingly, all were in a localized disease stage without distant metastases at the time of diagnosis. Immunohistochemical analysis showed expression of synaptophysin, chromogranin, GATA3, and nuclear hormone receptors (estrogen receptor (ER), progesterone receptor (PR)). Ki-67 was below 10% in all cases. All patients underwent partial mastectomy including lymphadenectomy. Two patients received radiation therapy, one of them received additional hormone therapy, while none of them were treated with chemotherapy. All remained free from recurrence during follow-up.

Our patients with small cell carcinoma (cases 4 and 5) displayed clinical symptoms and underwent more extensive diagnostics including bone scintigraphy, bronchoscopy, colonoscopy, and upper endoscopy. Moreover, those patients were in a more advanced disease stage with distant metastases already at the time of diagnosis. Histological analysis showed poor differentiation, featuring a Ki-67 proliferation index of at least 40%. Immunohistochemical staining revealed positive expression of synaptophysin in both cases and additionally weak expression of chromogranin in one case. There was no expression of the estrogen receptor (ER) in both cases, while case 4 at least showed weak expression of the progesterone receptor (PR). HER2neu was negative in both cases. In case 5 the transcription factor GATA3 was indicative of a primary tumor of the breast, while in case 4 E-cadherin was found. Unfortunately, additional staining for transcription factors (e.g., GATA3) in case 4 was not possible, since there was not enough material left and the patient is deceased. Both patients with small cell carcinoma of the breast initially received chemotherapy with Carboplatin and Etoposide but died after a rather short period of tumor response.

### 3.2. Terminology, Frequency, Epidemiology

Primary neuroendocrine carcinomas of the breast are rare and were first described by Feyrter and Hartmann in 1963 [21]. In 2002, Sapino et al. first proposed a more specific definition for NECB [22], which was subsequently adopted by the World Health Organization (WHO) as a unique type of breast cancer the following year [17]. The incidence of NECB is only poorly understood [13,15,16,17]. Lopez et al. [16] and Günhan et al. [15] analyzed 1368 and 1845 cases of breast cancer, respectively, and found that, by using WHO diagnostic criteria of 2003, NECB only account for 0.3% and 0.5% of breast cancer, which is much less than the 2–5% rate reported by the WHO [17]. In this context, it is important to note that NECB might be under-reported since NET specific markers are not routinely applied in breast cancer patients and many NECB might been overseen in clinical practice. According to the WHO Classification of Tumors of the Breast of 2012, NECB are stratified into three subgroups based on morphology: (1) well-differentiated neuroendocrine tumors, (2) invasive carcinomas of the breast with neuroendocrine differentiation, and (3) small cell carcinomas of the breast [17]. As opposed to the previous guidelines of 2003, diagnosis can be set up regardless of the percentage (50% threshold) of tumor cells expressing neuroendocrine markers [17]. Well-differentiated NECB (G1, G2) are characterized by a low proliferation index, retained expression of somatostatin receptors (SSTR), and are associated with an extraordinary favorable prognosis when compared to other malignancies. In contrast, small cell carcinomas of the breast are histologically poorly differentiated, feature a high Ki-67 proliferation index of >20%, and are associated with a dismal prognosis. Most patients with NECB are postmenopausal women and the incidence in males and younger women is low [23]. The median age of 64 years (49–78) in our case series is consistent with results from a Chinese case series including 126 cases with NECB [23] and previous reports [17,22]: only one patient (case 3) was diagnosed at age of 49 and was in perimenopause at that time.

### 3.3. Clinical Presentation and Diagnostic Work-Up

Diagnosis of NECB can only be made by the pathologist, since the clinical presentation and imaging findings are not distinct from other types of breast cancer. In many cases, NECB present as painless, palpable retro-areolar mass with secondary symptoms such as nipple retraction, fixation to deep tissues, skin ulceration, lymphadenopathy, or bloody nipple discharge [23,24]. In our case series, no patient suffered from bloody nipple discharge, carcinoid syndrome or hormonal hypersecretion and only four patients suffered from clinical symptoms such as pain, erythema, palpable mass, and skin retraction before initial diagnosis. NECB represent rather small tumors. In our case series, median tumor size was 18 mm (10–53 mm). In line, Lopez et al. reported sizes from 7 to 53 mm [16].

Radiological features of NECB are unspecific in most cases. However, some studies suggested that NECB might appear as a round, sharply circumscribed, hyperdense mass in mammography and as an irregular or microlobulated hypoechoic solid mass with increased vascularity on ultrasound [15,25,26]. Additionally, somatostatin receptor scintigraphy (SRS) or positron-emission tomography/computed tomography (PET/CT) with 68-Gallium-labeled somatostatin analogs (e.g., DOTATOC) may be used to detect well-differentiated NET, while 18-fluorodeoxyglucose (FDG)PET-CT can be used in poorly differentiated NEC with a high proliferation rate [27,28,29].

All patients in this study with well/moderately differentiated NECB received a punch biopsy after mammography and ultrasound of the breast. Further clinical work-up included CT-imaging and one case of Ga68-PET showing an SSR-positive primarius (case 1) before surgery. Patients with small cell carcinoma underwent more comprehensive diagnostics including CT-imaging, bone scintigraphy, bronchoscopy, colonoscopy, and upper endoscopy.

### 3.4. Histology

A reliable pathological diagnosis can only be made by analyzing specimens obtained by core needle biopsy or surgery, whereas fine-needle aspiration cytology is not recommended due to the similarity of NECB’s cytological features to those of invasive ductal carcinoma and intra-ductal papilloma [27,29,30]. Primary NECB comprise several histologic subtypes that differ from one another in cell type, level of invasiveness, and growth pattern [31]. Until 2003 WHO defined NECB as tumors of epithelial origin, with morphology similar to gastrointestinal and pulmonary neuroendocrine tumors and positive staining for neuroendocrine markers in at least 50% of the total cell population [17]. The following WHO classification guidelines 2012 do not include a percentage of tumor cells expressing neuroendocrine markers as the WHO acknowledged that the 50% threshold of cells with neuroendocrine marker expression was arbitrary [17,32]. In our experience, it is sometimes difficult to differentiate neuroendocrine morphology from a ductal morphology. This is exemplified in our patient no 3, who was initially diagnosed with DCIS. A threshold value of positive staining for neuroendocrine markers in at least 50% of the total cell population may be helpful in order to have objective criteria allowing different pathologists to come to the same conclusion in cases that are not entirely clear.

The utilization of detailed immunohistochemical staining and various imaging modalities are essential in establishing an accurate diagnosis. Neuroendocrine markers such as chromogranin and synaptophysin have the best sensitivity and specificity for immunohistochemical evaluation [33]. Other less specific markers showing positive expression are neuron-specific enolase (NSE), CK7, and CK56 [16,34]. TTF-1 and CDX2 stains are typically negative [35]. While NECB is often positive for hormone receptors (ER, PR), HER2neu is usually negative, although there are also reports of HER2neu-positive NECB [16,36]. In our case series, only one patient showed slight positivity of HER2neu expression. Expression of estrogen and progesterone receptors was demonstrated in all our cases with well and moderately differentiated NECB, while in one case of small cell carcinoma of the breast hormone receptor status was negative and in the other case PR was only expressed in 15% of tumor cells. In this context, Shin et al. report that hormone receptors in small cell neuroendocrine tumors are expressed in fewer cases (<70%) than in well/moderately differentiated NECB [37]. There are indications that these receptors are also expressed in neuroendocrine tumors of other origin. According to several expert groups PR expression in NET metastases indicates in many cases a pancreatic primary and can also be associated with duodenal origin [38,39,40]. Further, Curioni-Fontecedro et al. demonstrated that ER expression is frequent in all NEN of the lungs [41]. Hence, a prospective single-arm, unicentric clinical trial (HORMONET, NCT03870399) testing tamoxifen in well differentiated gastroenteropancreatic and pulmonal NET has been started [42].

Due to the architectural similarity to neuroendocrine tumors from other origin, NECB can be easily mistaken for neuroendocrine tumors metastatic to the breast. Finding a DCIS component indicates the breast as a primary tumor location [43]. The expression of transcription factors such as GATA3, which is a more sensitive and specific marker than mammoglobin, hints towards a mammary origin. Despite GATA3 also being expressed in urothelial, renal, germ cell tumors, and paragangliomas [44,45,46], a positive expression in extramammary NEN has not been described [43,47]. Additionally, positive expression of hormone receptors (PR, ER) in well/moderately differentiated NECB also play an important role in differentiating primary and secondary lesions [43,47,48]. In our case series, GATA3 was positive in all three cases, in which it was performed, while a DCIS component was described only in one patient (case 3). Transcription factors such as TTF-1 and CDX2, which show positivity in metastases of the lung (TTF1) and of gastrointestinal origin (CDX2), were negative [29,47]. All our patients with well and moderately differentiated NECB showed positive hormone receptor expression. As hormone receptors are expressed in fewer cases in small cell carcinomas of the breast in contrast to NECB [37], the presence of PR in case 4 is indicative of a primary of the breast. Additionally, in a series of 18 metastatic NETs in the breast by Perry et al. no expression of PR was found, differentiating these metastatic lesions from primary breast tumors [49].

### 3.5. Management

As there is a lack of large clinical studies, there are almost no standardized recommendations for treatment of NECB. Most treatments of NECB reported in the literature and in the present study (only in regard to well/moderately differentiated NECB) are similar to the treatment of ductal-type, while Anlauf et al. highlight the importance of treatment according to NET guidelines [20]. According to both guidelines, surgery is the mainstay of treatment for early NECB. The surgical procedure (breast conserving partial mastectomy, total mastectomy) depends on the location of the tumor and the clinical stage [50,51]. In this context, patients in cases 1–3 underwent partial mastectomy, while patients with small cell neuroendocrine carcinoma were not administered to surgery due to widespread metastases. In well/moderately differentiated NECB surgery is usually followed by radiotherapy depending on the size of the tumor and lymph node status [27,29]. In line, two out of three patients were subjected to radiotherapy. Chemotherapy is used as adjuvant therapy in patients with high risk of relapse or as neoadjuvant therapy in cases of locally advanced or inoperable NECB [29]. Combinations of platinum agents and etoposide, as it is recommended for small cell neuroendocrine tumors, and taxane-based chemotherapy, routinely used for other types of breast cancer, are commonly administered [30,52]. Both of our patients with small cell carcinoma of the breast initially received chemotherapy with Carboplatin and Etoposide, which is first-line therapy according to NET treatment guidelines [53]. Nevertheless, Inno et al. recommend treatment according to guidelines on ductal carcinoma due to lack of data for Cisplatin/Etoposide in breast tumors.

Antihormonal therapy has proven efficacy in patients with hormone receptor-positive breast carcinomas. According to Richter-Ehrenstein et al. adjuvant antihormonal therapy is the standard adjuvant therapy in hormone receptor positive NECB [48], which is why all patients with NECB (*n* = 7) in Lopez et al.’s study received antihormone therapy [16]. In contrast, only one of our patients with NECB and positive hormone receptor expression received antihormonal therapy as the other two patients rejected treatment due to lack of established guidelines. The prognostic role of HER2neu in NECB is not clear, but it can be assumed that it is analog to other invasive breast carcinomas, therefore anti-HER2neu therapy can also be considered for HER2neu-positive NECB. In the case of SSR-positive tumors, peptide receptor radionuclide therapy (PRRT), which is a tumor-targeted systemic radiotherapy that enables the specific delivery of radionuclides directly to tumor cells inducing tumor cell death [54,55,56], has been recommended after failure of conventional chemotherapy or also as first or second line therapy [20,57].

Thus, optimal treatment of NECB requires simultaneous consideration of both neuroendocrine and non-neuroendocrine breast tumor features. Nevertheless, a majority of expert groups recommend mainly treatment according to ductal carcinoma guidelines as a consequence of scarcity of available data. This clearly indicates the need for further studies to investigate sustainability. In our opinion, it is particularly useful to consider NET guidelines with regard to positive SSR status and possible diagnostic and therapeutic modalities.

## 4. Patients and Methods

We screened our database of 612 patients with histologically proven diagnosis of NEN from 2008 to 2019 for cases of neuroendocrine carcinoma of the breast. Our database includes information on primary and metastatic tumor localizations, histology, including mitotic rate or Ki67 proliferation index, diagnostic methods used for detection of primary and metastatic tumors, classification according to Capella, WHO, and TNM staging and grading, among others. Only 12 patients were found in total in our NEN database, in which the breast was recorded to be affected by the disease. We excluded seven patients that did not fulfil criteria of NECB. In five cases breast carcinoma with partial neuroendocrine differentiation rather than primary neuroendocrine carcinoma was present according to expert pathology assessment based on WHO guideline criteria 2012, which defines NECB as tumors of epithelial origin, with morphology similar to gastrointestinal and pulmonary neuroendocrine tumors and positive staining for neuroendocrine markers. As there is no longer a numeric threshold included in the WHO criteria of 2012, the decision on whether a primary neuroendocrine neoplasm or a tumor with only partial neuroendocrine differentiation is present, depends more on the expert pathologist’s judgement. Two additional cases had to be excluded because they were lost to follow-up and histologic specimens could not be retrieved for reevaluation. The remaining five cases displayed a histologically confirmed NECB and had a complete clinical follow-up at our institution.

For literature review, PUBMED was searched using a combination of the following keywords: neuroendocrine carcinoma of the breast, small cell carcinoma of the breast, clinicopathological features, histology, immunohistochemical profiling, imaging, management, prognosis, molecular markers. Published literature was reviewed with respect to demographic data (age, sex) as well as clinical features including metastases, symptoms, complications, treatment, and diagnostic methods.

## 5. Conclusion and Future Perspectives

Primary neuroendocrine carcinoma of the breast represents a rare, but probably under-diagnosed, entity. Within this study, we analyzed the clinical course of these patients and highlighted the diagnostic and therapeutic management of these patients. Current strategies in the diagnosis and therapy of NECB rely on radiology and conventional histology as well as on cytotoxic chemotherapy and somatostatin analogs, respectively. However, given the variable and often long-lasting clinical course of NECB as well as the upcoming molecular targeted therapies, it is mandatory to discuss each individual case in an interdisciplinary group of NEN experts to provide each individual patient with the optimal treatment. Areas that deserve specific attention and work in this context are those that link management and therapy to profiling at all given omics levels. This may be facilitated through a wide application of high throughput technologies made available to the bedside.

## Figures and Tables

**Figure 1 cancers-12-00733-f001:**
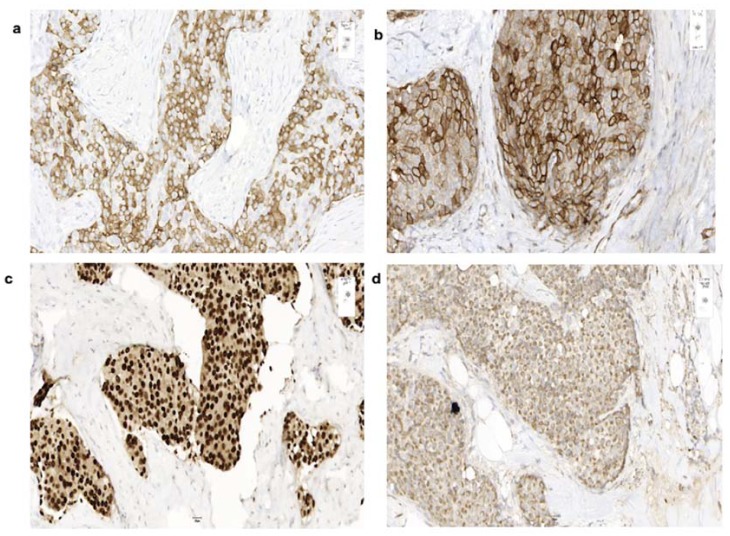
Immunohistochemical expression of cells of the resected tumor of a 73-year-old patient with a well-differentiated neuroendocrine tumors (NET) of the breast. Immunohistochemical staining (20×) shows a well-differentiated neuroendocrine tumor with positivity for synaptophysin (**a**), SSTR2A (**b**), estrogen receptors (**c**), and GATA3 (**d**).

**Figure 2 cancers-12-00733-f002:**
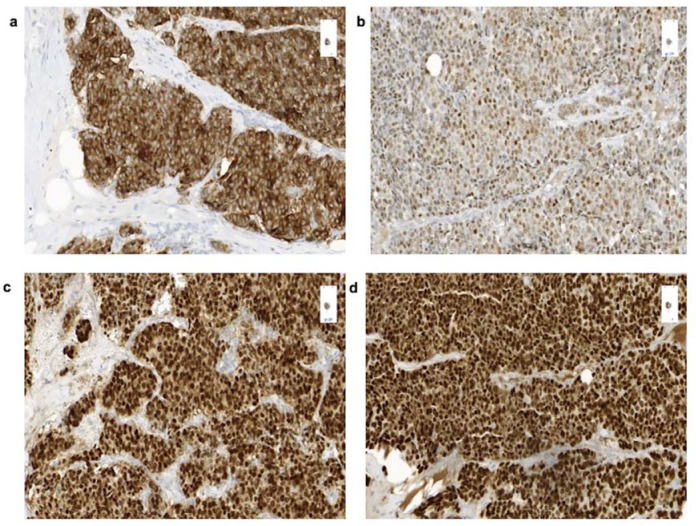
Immunohistochemical expression of cells of the resected tumor of a 53-year-old patient. with a well-/moderately differentiated NET of the breast. Immunohistochemical staining (20×) reveals positivity for synaptophysin (**a**), progesterone receptors A (**b**), estrogen receptors (**c**), and GATA3 (**d**).

**Figure 3 cancers-12-00733-f003:**
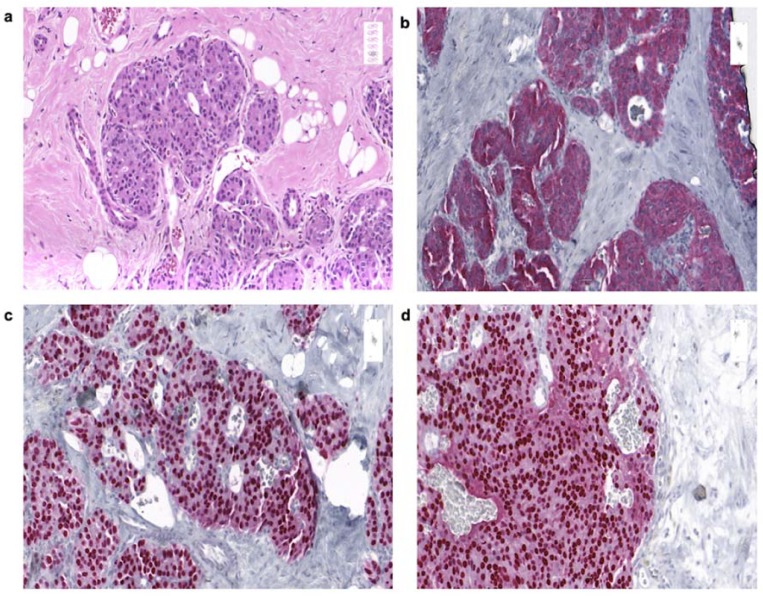
Pathohistological analysis and immunohistochemical expression of cells of the resected tumor of a 49-year-old patient with moderately differentiated NECB. H&E stain (20×) shows ductal carcinoma in situ (DCIS) with cells of cribriform architecture, nuclei with moderate pleomorphism, and abundant eosinophilic cytoplasm (**a**). Immunohistochemical staining (20×) reveals positivity for synaptophysin (**b**), estrogen receptors, and (**c**) progesterone receptors (**d**).

**Figure 4 cancers-12-00733-f004:**
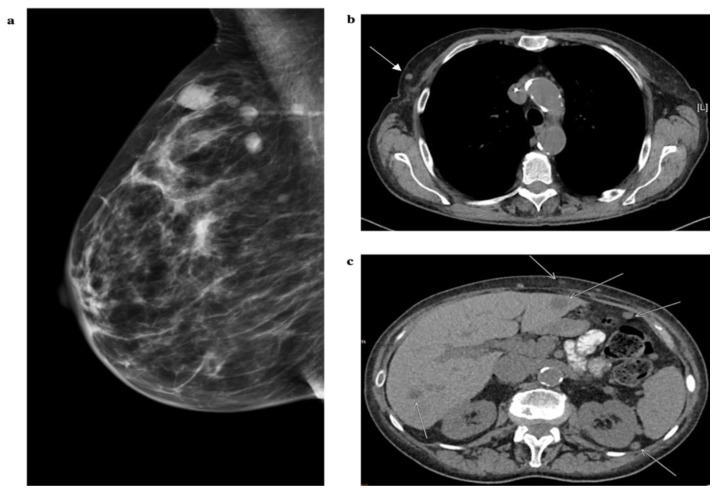
Mammography and axial computed tomography (CT)-scans of a 78-year-old female patient with small cell carcinoma of the breast and multiple metastases in liver, bone, pleura, and peritoneal carcinomatosis. (**a**) Mammography of the right breast demonstrates multiple round and oval high-density masses with indistinct margin. (**b**) Axial CT scan of the thorax shows an oval mass in the right breast indicated by the white arrow. (**c**) Axial CT scan of the abdomen displays multiple hepatic metastases, peritoneal carcinomatosis, and subcutaneous metastases indicated by white arrows.

**Figure 5 cancers-12-00733-f005:**
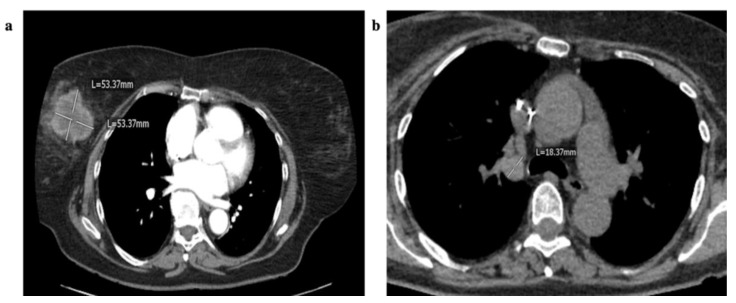
Axial CT-scans of a 67-year-old female patient diagnosed with a neuroendocrine carcinoma (NEC) G3 of the right mamma and synchronous liver and lymph node metastases. (**a**) Axial contrast-enhanced CT scan in the arterial phase (15 s after contrast injection) shows a mass in the right breast with rim enhancement. (**b**) Contrast-enhanced axial CT scan in the venous phase (75 s) reveals enlarged right-sided hilar lymph nodes.

**Figure 6 cancers-12-00733-f006:**
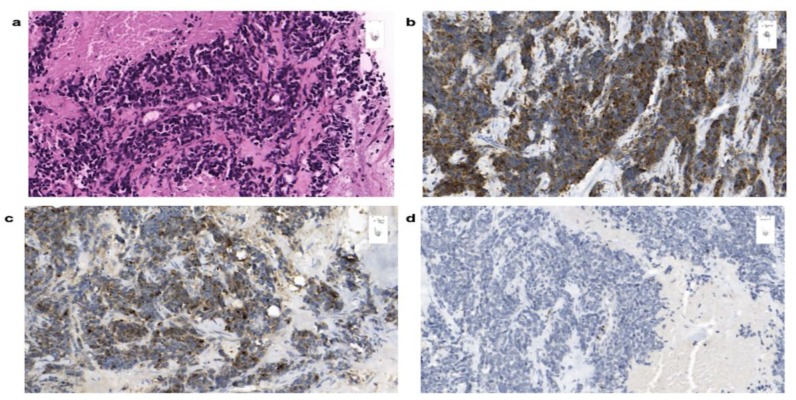
Pathological analysis and immunohistochemical expression of cells of obtained biopsy of a 67-year-old patient with poorly differentiated small cell carcinoma of the breast. H&E stain (20×) shows small nuclei, dense chromatin, and scant cytoplasm (**a**). Immunohistochemical staining (20×) reveals positivity for synaptophysin (**b**), chromogranin A (**c**), and GATA3 (**d**).

**Table 1 cancers-12-00733-t001:** Clinical features of five patients with neuroendocrine carcinoma of the breast (NECB).

Patient no.	Sex	Age	Date of Diagnosis	Diagnosis	Breast Localization	Family History	Primary Tumor Size (Diameter)	Axillary Node Status	Distant Metastases
1	F	73	Feb-19	Well-differentiated NET of the breast	Inferior-external, left	Yes (breast cancer)	10 mm	1/8	None
2	F	53	Nov-17	Well-/moderately-differentiated NET of the breast	Inferior, right	No	12 mm	0/3	None
3	F	49	Mar-09	Moderately-differentiated NET	Upper external, right	Yes (breast cancer)	15 mm	1/0	None
4	F	78	May-18	Poorly differentiated small cell neuroendocrine carcinoma	Upper-external, right, upper external left	Yes (small cell lung cancer)	Multiple lesions up to 9 mm	N.a.	Lymph nodes, bones, liver, pleura, peritoneum
5	F	67	Sep-16	Poorly differentiated small cell neuroendocrine carcinoma	Upper, right	No	53 mm	N.a.	Lymph nodes, liver

**Table 2 cancers-12-00733-t002:** Histological and immunohistochemical features of five patients with NECB.

Patient no.	Grading	Ki67	Intrinsic Subtype	Immunohistochemistry	Transcription Factors
1	G2	6%	100% ER, 40% PR, Her2neu-	Synaptophysin+, chromogranin slightly+, CK-MNF116+, CK18+, SSTR2A+	GATA3+, TTF-1-, CDX-2-, Islet1-
2	G2	<5%	>80% ER, >90% PR, Her2neu-	Synaptophysin+, chromogranin slightly + CK18+, SSTR2A+	GATA3+, TTF-1-, CDX-2-, Islet1-
3	G2	5%	100% ER, 100% PR, Her2neu +1	Synaptophysin+, chromogranin+, CK5/6-CK14-	N.a.
4	G3	47%	0%ER, 15% PR, Her2neu-	Synaptophysin+, chromogranin-, MNF116+, CD 56 +, AE1/3+, E-cadherin+, CD3-, CD20- BCL2-	N.a.
5	G3	80%	0%ER, 0%PR, Her2neu-	Synaptophysin+, chromogranin+, MNF116+	GATA3+, TTF-1-, CDX2-

**Table 3 cancers-12-00733-t003:** Treatment schedules and follow-up of five patients with NECB.

Patient no.	Surgery	Radiotherapy	Antihormone Therapy	Chemotherapy	Follow-Up 01/2020
1	Partial mastectomy, lymphadenectomy, R0	No	No	No	Alive, CR
2	Partial mastectomy, sentinel lymph node resection, R0	Yes (15cycles)	Yes	No	Alive, CR
3	Partial mastectomy, sentinel lymph node resection, R0	Yes	No	No	Alive, CR
4	No	No	No	06/2018–02/2019 Carboplatin (300/m^2^)/Etoposide(120/m^2^)	Death Feb-2019
5	No	No	No	09/16–12/16: 5 cycles Cisplatin (80/m^2^) Etoposide (120/m^2^) 03/17 2nd line with Doxorubicin (16 mg/m^2^), Cyclophosphamide (750 mg/m^2^), Vincristine (0.4 mg/m^2^) in a 80% reduced dose	Death Jul-2017

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
