# Peer review of "Primary Neuroendocrine Neoplasms of the Breast: Case Series and Literature Review"

_cancers, 2020, doi:10.3390/cancers12030733_

Round 1

Reviewer 1 Report

Özdirik et al. report on a case series of 5 patients with primary neuroendocrine neoplasms of the breast and review the current literature. Three of their five patients have been diagnosed with non-metastatic well differentiated neuroendocrine carcinoma of the breast (NECB), while two have been diagnosed with disseminated poorly-differentiated small cell carcinoma. The authors provide the clinical background, diagnostic work-up, treatment and outcome of these patients. Within the literature review the authors state current case definitions according to WHO 2012, explore the reported and predicted incidence of NECB and discuss current diagnostic work-up and management.

Overall the review is well written and informative to the reader with a special interest in neuroendocrine tumors. However some points need further attention.

  1. The definition of case 4 with disseminated lesions in the breast as a primary SC-NEC of the breast is not as clear as stated by the authors. What favored the diagnoses of primary NECB over metastatic lesions in the breast in this case?
  2. Differential diagnosis for primary NECB versus metastatic lesions to the breast from an unknown primary should be more thoroughly discussed.
  3. The authors do not discuss ER / PR positivity in NET from other primaries into detail. Similarly, the diagnostic role of GATA3 is not clearly stated. What is known about GATA3 in primary and metastatic NET?
  4. The exclusion of 5 cases (“did not fulfil criteria according to expert pathology assessment”) should be stated in more detail, as this might be informative for the reader as well.
  5. Provision of histology and immunohistochemistry would be helpful to illustrate cases.

Author Response

Reviewer: 1

Overall the review is well written and informative to the reader with a special interest in neuroendocrine tumors. However some points need further attention.

Response: We are grateful to the reviewer for his positive feed-back and for carefully reading our manuscript. By answering on all of the reviewers´ questions, we hope that we were able to significantly improve the manuscript.

Specific comments:

1.The definition of case 4 with disseminated lesions in the breast as a primary SC-NEC of the breast is not as clear as stated by the authors. What favored the diagnoses of primary NECB over metastatic lesions in the breast in this case?

Response:

This is certainly a very important comment. The diagnosis of small cell neuroendocrine carcinoma in case of number 4 was made under consideration of clinical and pathological findings. Furtherclinicalwork-upincludingbronchoscopy,upperendoscopy,andcolonoscopydidnotprovideanyevidenceofanotherpossibleprimarytumor. The stained tissue slides of the resected tumor were reviewed, hinting towards a small cell carcinoma of the breast.  Immunohistochemical analysis revealed positive expression of synaptophysin, E-cadherin and progesterone receptors in 15% of tumor cells. As hormone receptors are expressed in fewer cases in small cell carcinomas of the breast in contrast to NECB[37], the presence of PR in this case is indicative of a primary of the breast. Furthermore, positive e-cadherin expression hints towards a mammary primary, which is expressed in primary breast tumors[19,20]. We would have favored to perform further immunohistochemical analysis including transcription factors such as GATA3, TTF and CDX2 to support our diagnosis, but unfortunately, there is not enough material for further staining and no additional biopsy is possible anymore as the patient is dead already.

 “ Bronchoscopy,upperendoscopy,andcolonoscopydidnotprovideanyevidenceofanotherpossibleprimarytumor. A core needle biopsy from the right breast was taken, showing a small cell carcinoma of the breast.Immunohistochemicalstainingrevealedpositive expressionofsynaptophysin and E-cadherin, which is expressed in primary breast tumors[19,20].Nuclear PR expression was positive in 15% of tumor cells.”(page6, l.180)

“As hormone receptors are expressed in fewer cases in small cell carcinomas of the breast in contrast to NECB[37], the presence of PR in case 4 is indicative of a primary of the breast. Also, in a series of eighteen metastatic NETs in the breast by Perry et al. no expression of PR was found, differentiating these metastatic lesions from primary breast tumors[49].” (page 12, l.375)

  1. Differential diagnosis for primary NECB versus metastatic lesions to the breast from an unknown primary should be more thoroughly discussed.

Response: We agree with the reviewer on this point. Therefore, we included a new paragraph, in which we pointed out the histopathological differences between NECB and breast metastases.

“Due to the architectural similarity to neuroendocrine tumors from other origin, NECB can be easily mistaken for neuroendocrine tumors metastatic to the breast. Finding a DCIS component indicates the breast as a primary tumor location[43]. The expression of transcription factors such as GATA3, which is a more sensitive and specific marker than mammoglobin, hints towards a mammary origin. Despite GATA3 also being expressed in urothelial, renal, germ cell tumors and paragangliomas[44-46], a positive expression in extramammary NEN has not been described [43,47].Additionally, positive expression of hormone receptors (PR, ER) in well/moderately differentiated NECB also play an important role in differentiating  primary and secondary lesions [43,47,48]. In our case series, GATA3 was positive in all three cases in which it was performed, while a DCIS component was described only in one patient (case 3). Transcription factors such as TTF-1 and CDX2, which show positivity in metastases of the lung (TTF1) and of gastrointestinal origin (CDX2), were negative[29,47]. All our patients with well and moderately differentiatedNECB showed positive hormone receptor expression.  As hormone receptors are expressed in fewer cases in small cell carcinomas of the breast in contrast to NECB[37], the, albeit weak, presence of PR in case 4 is indicative of a primary of the breast. Also, in a series of eighteen metastatic NETs in the breast by Perry et al. no expression of PR was found, differentiating these metastatic lesions from primary breast tumors[49].”  (p.11, l.362)

  1. The authors do not discuss ER / PR positivity in NET from other primaries into detail. Similarly, the diagnostic role of GATA3 is not clearly stated. What is known about GATA3 in primary and metastatic NET?

Response: This is certainly a very important comment. Therefore, we included a paragraph discussing hormone receptor expression in other neuroendocrine primaries as well as a paragraph describing histomorphological differences between NECB and metastases in which we also elucidate the role of GATA3 and hormone receptors .

  1. The exclusion of 5 cases (“did not fulfil criteria according to expert pathology assessment”) should be stated in more detail, as this might be informative for the reader as well.

Response: This is certainly a very important comment. Therefore, we included a paragraph discussing more in detail why the other seven patients could not be included into our study.

“Only12patientswerefoundintotalinourNENdatabase,inwhichthebreastwasrecordedtobeaffected bythedisease.We excluded sevenpatientsthatdidnotfulfilcriteriaofNECB.Infivecasesbreastcarcinomawithpartial neuroendocrinedifferentiationratherthanprimaryneuroendocrinecarcinomawaspresentaccordingtoexpertpathologyassessment based on WHO guideline criteria 2012, which defines NECB as tumorsofepithelialorigin,withmorphologysimilartogastrointestinalandpulmonaryneuroendocrinetumorsandpositivestainingforneuroendocrinemarkers.As there is no longer a numeric threshold included in the WHO criteria of 2012, the decision on whether a primary neuroendocrine neoplasm or a tumor with only partial neuroendocrine differentiation is present, depends more on the expert pathologist’s judgement. Two additionalcaseshadtobeexcludedbecausetheywerelosttofollow-upandhistologicspecimenscouldnotberetrievedforreevaluation. TheremainingfivecasesdisplayedahistologicallyconfirmedNECBandhadacompleteclinicalfollow-upatourinstitution.”(p.13; l.430)

  1. Provision of histology and immunohistochemistry would be helpful to illustrate cases.

Response: We included several figures showing pathohistological and immunohistochemical analysis. If desired, these figures can be taken out again. Addtionally, we have taken out (previous) figure 1 and replaced it with images including further immunohistochemical analysis.

Reviewer 2 Report

This is a comprehensive review on a very rare tumour entity: neuroendocrine tumour of the breast. Authors describe 5 cases observed in their own institution and add a review on the available lterature. I would recommend acceptance for publication. Before that I have a view questions to the authors:

1) Introduction, 2nd paragraph: " with a low risk of distant metastases". Please, delete this phrase since G1 NEts of the midgut produce frequently local and distant metastases.

2) I miss in cases 1-3 a more detailed pathological work-up. Authors only write that a reference pathologist found  a " well-differentiated neuroendocrine tumor". I accept that these tumours were NENs but the question is whether or not some tumour cells were  of ductal origin in the sense of a ductal carcinoma. I doubt that these tumours were pure NENs.

3) You found in most tumours estrogen and progesteron receptors and in one tumour Her2neu expression: please indicate the location of these receptors.

4) Patient 3 received antihormonal therapy. Patient 1 and 3 not. Give more details on the indication for or against antihormonal therapy.

5) "The review of the literature and discussion" is too lengthy and can be shorted by at least 30%

Author Response

Reviewer: 2

Comments to the Author

This is a comprehensive review on a very rare tumour entity: neuroendocrine tumour of the breast. Authors describe 5 cases observed in their own institution and add a review on the available literature. I would recommend acceptance for publication. Before that I have a view questions to the authors:

Response: We are grateful to the reviewer’s positive feed-back and the recommendation for acceptance for publication.  By answering all questions, we hope that we were able to provide the required clarifications.

Specific comments:

  1. Introduction, 2nd paragraph: " with a low risk of distant metastases". Please, delete this phrase since G1 NEts of the midgut produce frequently local and distant metastases.

Response: We appreciate the reviewer’s comment. We have removed the sentence “These tumors tend to follow a prolonged clinical course with a low risk of distant metastases.”  from the manuscript, as kindly suggested.

  1. I miss in cases 1-3 a more detailed pathological work-up. Authors only write that a reference pathologist found a " well-differentiated neuroendocrine tumor". I accept that these tumours were NENs but the question is whether or not some tumour cells were of ductal origin in the sense of a ductal carcinoma. I doubt that these tumours were pure NENs.

Respond: Thank you for your important comment. Prior to inclusion in our study, tissue samples of all patients in our NEN database, in which the breast was recorded to be affected by the disease, and whose permission could be obtained, were reanalyzed by our expert pathologist according to WHO criteria of 2012. We explicitly asked our expert pathologist for the presence of DCIS component, which was only found in one case. In all cases (1-3) synaptophysin and chromogranin were strongly positive. Transcription factors indicating a non-mammarian primary such as TTF and CDX2 were negative. GATA 3 was positive in two of three cases and SSTR2A expression was found to be positive in one case (Table 2.Histologicalandimmunohistochemicalfeaturesoffive patientsNECB; page 2, l.80)

  1. You found in most tumours estrogen and progesteron receptors and in one tumour Her2neu expression: please indicate the location of these receptors.

Response: We fully agree with the reviewer on this point. Therefore, we added in each case the location of the analysed nuclear estrogen and progesterone and membranous HER2Neu receptors (page 3, l.102; page 4 l.135; page 6 l.160; page7, l.195).

  1. Patient 2 received antihormonal therapy. Patient 1 and 3 not. Give more details on the indication for or against antihormonal therapy.

Response: Thank you for this important comment. The reason why patient 1 did not receive any additional adjuvant treatment is because she rejected recommended adjuvant therapy. In case of patient no. 3, the patient discussed antihormonal treatment with her primary care gynecologistafter adjuvant radiotherapy. Since there were no clear guidelines on antihormonal therapy for neuroendocrine neoplasia, the patient rejected treatment with tamoxifen.We included a paragraph in which we elucidate the indications for antihormonal therapy more precisely.

“Antihormonal therapy has proven efficacy in patients with hormone receptor-positive breast carcinomas. According to Richter-Ehrenstein et. al. adjuvant antihormonal therapy is the standard adjuvant therapy in hormone receptor positive NECB[48], which is why all patients with NECB (n=7) inLopez et al.’s study receivedantihormonetherapy[54].In contrast,only one of our patients with NECB and positive hormone receptor expression received antihormonal therapy as the other two patients rejected treatment due to lack of established guidelines.”(p.12,l.405)

After adjuvant radiotherapy the patient discussed antihormonal treatment with her primary care gynecologist and rejected further treatment with tamoxifen, since there were no clear guidelines on antihormonal therapy for NEN.(page 6,l.168). 

  1. "The review of the literature and discussion" is too lengthy and can be shorted by at least 30%

Response: We are grateful for this important comment. Therefore, we have shortened the review of the literature and discussion part.

Round 2

Reviewer 1 Report

Özdirik and Kayser et al. have convincingly addressed all concerns. In particular, the addition of histology slides and the discussion of PR/ER positivity in NET of other origin has greatly improved the manuscript.